# m6A modifications regulate intestinal immunity and rotavirus infection

Anmin Wang[1,2], Wanyin Tao[1,2], Jiyu Tong[3], Juanzi Gao[1], Jinghao Wang[1], Gaopeng Hou[4], Chen Qian[1], Guorong Zhang[1,2], Runzhi Li[1,2], Decai Wang[1,2], Xingxing Ren[1,2], Kaiguang Zhang[1], Siyuan Ding[4], Richard A Flavell[5,6], Huabing Li[3], Wen Pan[1,2]*, Shu Zhu[1,2,7,8]*

[1]Department of Digestive Disease, The First Affiliated Hospital of University of Science and Technology of China, Division of Life Sciences and Medicine, University of Science and Technology of China, Hefei, China; [2]Institute of Immunology, the Chinese Academy of Sciences Key Laboratory of Innate Immunity and Chronic Disease, Division of Life Sciences and Medicine, University of Science and Technology of China, Hefei, China; [3]Shanghai Institute of Immunology, Department of Microbiology and Immunology, Shanghai Jiao Tong University School of Medicine (SJTU-SM), Shanghai, China; [4]Department of Molecular Microbiology, Washington University School of Medicine in St. Louis, St. Louis, United States; [5]Department of Immunobiology, Yale University School of Medicine, New Haven, United States; [6]Howard Hughes Medical Institute, Yale University School of Medicine, New Haven, United States; [7]School of Data Science, University of Science and Technology of China, Hefei, China; [8]Institute of Health and Medicine, Hefei Comprehensive National Science Center, Hefei, China, Hefei, China

*For correspondence:
wenpan@ustc.edu.cn (WP);
zhushu@ustc.edu.cn (SZ)

Competing interest: The authors declare that no competing interests exist.

**Abstract** N6-methyladenosine (m6A) is an abundant mRNA modification that affects many biological processes. Nevertheless, we have a poor understanding of how m6A levels are regulated during physiological or pathological processes such as responses to virus infections. The in vivo function of m6A in intestinal immune defense against virus infections is largely unknown. Here, we describe a novel antiviral function of m6A modification during rotavirus (RV) infection in small bowel intestinal epithelial cells (IECs). We found that rotavirus infection induced global m6A modifications on mRNA transcripts by downregulating the m6a eraser ALKBH5. Mice lacking the m6A writer enzymes METTL3 in IECs (Mettl3ΔIEC) were resistant to RV infection and showed increased expression of interferons (IFNs) and IFN-stimulated genes (ISGs). Using RNA-sequencing and m6A RNA immuno-precipitation (RIP)-sequencing, we identified IRF7, a master regulator of IFN responses, as a primary target of m6A during virus infection. In the absence of METTL3, IECs showed increased Irf7 mRNA stability and enhanced type I and III IFN expression. Deficiency in IRF7 attenuated the elevated expression of IFNs and ISGs and restored susceptibility to RV infection in Mettl3ΔIEC mice. Moreover, the global frequency of m6A modifications on mRNA transcripts declined with age in mice, with a significant drop during the period from 2 weeks to 3 weeks after birth, which is likely to have broad implications for the development of the intestinal immune system against enteric viruses early in life. Collectively, our results demonstrate a novel host m6A-IRF7-IFN antiviral signaling cascade that restricts rotavirus infection in vivo.

## Editor's evaluation

This study clearly shows that m6A modification on mRNA regulates immune responses in the intestine during rotavirus infection. The authors further show the mechanisms how m6A modification is

regulated in the intestine. Thus, this study provides important insights into the regulation of anti-viral immunity in the intestine.

## Introduction

N6-methyladenosine (m6A) is the most abundant internal mRNA modification and modulates diverse cellular functions through m6A-related writers, erasers, and readers (*Roundtree et al., 2017*; *Xu et al., 2021*; *Chelmicki et al., 2021*). The m6A modification directly recruits m6A-specific proteins of the YT521-B homology (YTH) domain family (*Roundtree et al., 2017*). These proteins mediate the m6A-dependent regulation of pre-mRNA processing, microRNA processing, translation initiation, and mRNA decay (*Roundtree et al., 2017*). In recent works, m6A modifications, with either a pro-viral or anti-viral role, have been identified in the genomes of RNA viruses and in the transcripts of DNA viruses (*Tsai et al., 2018*; *Li et al., 2017*; *Hesser et al., 2018*; *Imam et al., 2018*; *Ye et al., 2017*). Furthermore, m6A RNA modification-mediated downregulation of the α-ketoglutarate dehydrogenase (KGDH)-itaconate pathway inhibits viral replication independently of the innate immune response (*Liu, 2019*). According to *Gao et al., 2020*, m6A modification preserves the self-recognition of endogenous transcripts. Deletion of the m6A writer *Mettl3* decreases the frequency of m6A modifications in endogenous retrovirus (ERV) transcripts. The accumulation of ERVs activates pattern recognition receptors (e.g. RIG-I) pathways, resulting in a detrimental interferon response in the livers of fetal mice (*Gao et al., 2020*).

m6A modification of the enterovirus 71 (EV71) RNA genome is important for viral propagation, and EV71 infection increases the expression of m6A writers in vitro (*Hao et al., 2019*). m6A methyltransferase METTL3 knockdown reduces EV71 replication, whereas m6A demethylase knockdown of the Fat mass and obesity-associated protein (FTO) increases EV71 replication (*Hao et al., 2019*). In addition, human cytomegalovirus can upregulate the expression of m6A-related proteins (*Rubio et al., 2018*; *Winkler et al., 2019*). Despite the information about m6A regulation and function during viral infection revealed by these in vitro studies, the regulation of m6A modification and the specific role of m6A in the antiviral response in vivo, especially in the gastrointestinal tract, remain unclear.

Rotavirus (RV), a member of the family *Reoviridae*, is a nonenveloped icosahedral-structured virus comprising 11 segments of double-stranded RNA. Children under the age of five are at high risk of rotavirus infection, which can cause severe diarrhea, dehydration, and death (*Crawford et al., 2017*). Rotaviruses encode multiple viral proteins that inhibit innate immune responses by degrading interferon regulatory factors (IRFs) and mitochondrial antiviral-signaling protein (MAVS), thus facilitating efficient virus infection and replication (*Ding et al., 2018*; *Barro and Patton, 2007*). The timely induction of an IFN response is key to the host's success in controling invading viruses, including RV (*Honda et al., 2005*; *Lin et al., 2016*; *Pott et al., 2011*). We found that rotavirus infection induced global m6A modifications on mRNA transcripts by downregulating the m6A eraser ALKBH5. Mice that lacked the m6A writer enzyme METTL3 in IECs (*Mettl3*ΔIEC mice) were resistant to RV infection. We identified IRF7, a master regulator of IFN responses (*Honda et al., 2005*), as one of the primary m6A targets during virus infection. In the absence of METTL3, IECs showed increased *Irf7* mRNA stability and enhanced expression of type I and III IFN. Deficiency in IRF7 attenuated the elevated expression of IFNs and ISGs and restored susceptibility to RV infection in *Mettl3*ΔIEC mice. Collectively, our results identified a novel regulation and function of m6A modifications in an enteric viral infection model in vivo.

## Results

### The regulation and function of mRNA m6A modifications during RV infection

RV infections primarily occur in humans in children under the age of five, and in mice in neonatals younger than 2 weeks old (*Crawford et al., 2017*; *Du et al., 2017*). Intriguingly, the frequency of total RNA m6A modifications in mouse ileal tissues, revealed by a m6A dot blot and mass spectrum (MS) analysis, significantly declined from 2 weeks to 3 weeks post birth (*Figure 1a, b and c*); this was the result of increased *Alkbh5* expression (*Figure 1d*, *Figure 1—figure supplement 1a* and

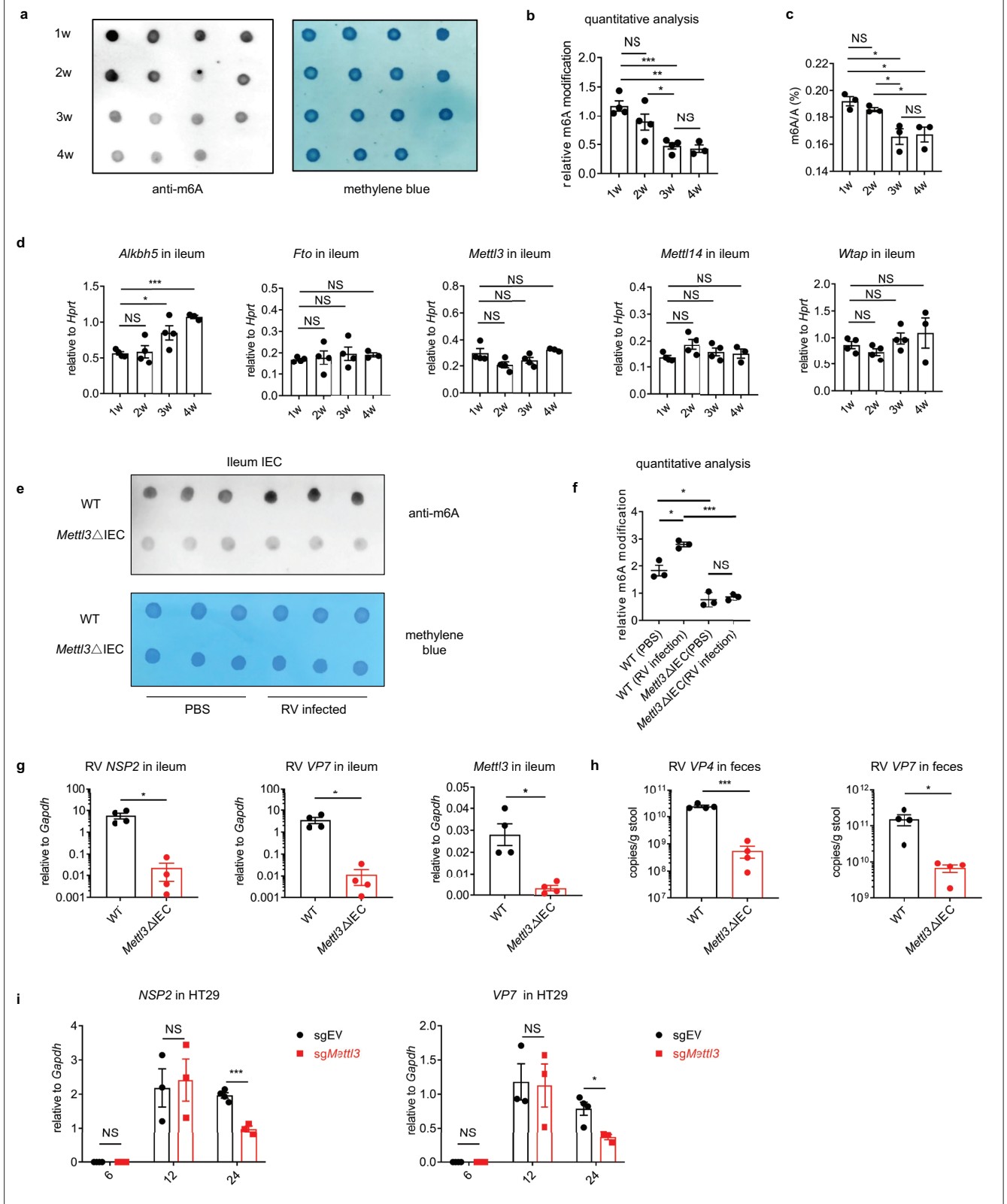

**Figure 1.** Rotavirus infection increases the global frequency of m6A modifications, and METTL3 deficiency in intestinal epithelial cells (IECs) results in increased resistance to rotavirus infection. (**a**) m6A dot blot analysis of total RNA in ileum tissues from mice of different ages. Methylene blue (MB) staining was used as the loading control. (**b**) Quantitative analysis of the dot blot analysis shown in (**a**) (mean ± SEM), statistical significance was determined by Student's t-test (*p < 0.05, **p < 0.005, ***p < 0.001, NS, not significant). The quantitative m6A signals were normalized against the

*Figure 1 continued on next page*

Figure 1 continued

quantitative MB staining signals. (**c**) Mass spectrum (MS) analysis of m6A level in ileum tissues from mice of different ages (mean ± SEM). Statistical significance was determined by Student's t-test (*p < 0.05, NS, not significant). (**d**) qPCR analysis of the expression of the indicated genes (relative to the expression of the reference gene *Hprt*) in ileum tissues from mice of different ages (mean ± SEM). Statistical significance was determined by Student's t-tests between groups (*p < 0.05, ***p < 0.001, NS, not significant). (**e**) WT and *Mettl3*ΔIEC mice were infected by rotavirus EW strain at 8 days post birth. m6A dot blot analysis of total RNA in ileum IEC at 2 days post infection (dpi). Methylene blue (MB) staining was used as the loading control. (**f**) Quantitative analysis of the dot blot analysis shown in (**e**) (mean ± SEM). Statistical significance was determined by Student's t-test (*p < 0.05, ***p < 0.001, NS, not significant). The quantitative m6A signals were normalized against the quantitative MB staining signals. (**g–h**) *Mettl3*ΔIEC mice and littermate controls were infected by the rotavirus EW strain at 8 days post birth. qPCR analysis of RV viral loads in ileum tissue (**g**) or fecal samples (**h**) from *Mettl3*ΔIEC mice and littermate controls was carried out at 4 dpi (littermate WT n = 4, *Mettl3*ΔIEC n = 4, mean ± SEM). Statistical significance was determined by Student's t-tests between genotypes (*p < 0.05). (**i**) qPCR analysis of the indicated genes in Rhesus rotavirus (RRV)-infected HT-29 cells transduced with *Mettl3* single guide RNA (sgRNA) or control sgRNA, at the indicated hours post infection (hpi) (mean ± SEM). Statistical significance was determined by Student's t-test (*p < 0.05, ***p < 0.001, NS, not significant). Experiments in (**a, d-f, and i**) were repeated twice, whereas those in (**g and h**) were are repeated four times.

The online version of this article includes the following figure supplement(s) for figure 1:

**Figure supplement 1.** ALKBH5 regulates total RNA m6A modification level in the intestine.

**Figure supplement 2.** Dot blot analysis of total RNA m6A modification level in the ileum of germ-free mice.

**Figure supplement 3.** Mass spectrum (MS) analysis of total RNA m6A modification level in RV-infected ileum.

**Figure supplement 4.** Schematic design of the RV infection model.

*Figure 1—figure supplement 1b*) and possibly also related to the development of the microbiota (*Figure 1—figure supplement 2*). Indeed, overexpression of *Alkbh5* in a mouse IEC cell line caused decreased m6A level (*Figure 1—figure supplement 1c*), supporting the role of *Alkbh5* in regulating global m6A levels in the intestine. These data indicate a potential link between resistance to RV infection and the declining frequency of m6A modifications during early-life development.

In addition, the global levels of m6A RNA modifications increased in the ileum tissue of suckling mice subsequent to infection with the RV murine strain EW (*Figure 1e and f* and *Figure 1—figure supplement 3*).By contrast m6A modification was barely detected in control IECs that had a deficiency in the m6A writer METTL3 (*Mettl3*^fl/fl^*Vil1*^Cre^, *Mettl3*ΔIEC) (*Figure 1e and f*). Thus, we hypothesize that RV may induce an enriched cellular m6A modification environment and a weakened innate immune response, thereby facilitating virus replication. To investigate the in vivo role of m6A in anti-RV immunity, we infected *Mettl3*ΔIEC mice and wild-type (WT) littermates with the RV EW strain. The viral RNA load in the ileum tissue of *Mettl3*ΔIEC mice was significantly lower than that in the same tissue from WT mice (*Figure 1g* and *Figure 1—figure supplement 4*). Fecal virus shedding was also significantly lower in *Mettl3*ΔIEC mice than in WT mice (*Figure 1h*). Genetic knockdown of METTL3 in HT-29 cells (a human colonic epithelial cell line) by CRISPR-mediated gene silencing also led to reduced RV replication (*Figure 1i*), further highlighting the resistance to RV infection phenotype caused by METTL3 deficiency.

## METTL3 deficiency in IECs results in decreased m6A deposition on *Irf7*, and increased interferon responses

To dissect the mechanism underlying the observed RV infection phenotypes, we performed RNA-sequencing using IECs from METTL3-deficient mice and littermate controls at steady state. Gene ontology analysis revealed that most of the genes that were expressed differentially in METTL3-deficient IECs vs WT IECs were enriched in the 'defense response to virus', 'response to interferon-beta', and 'positive regulation of innate immune response' pathways (*Figure 2a*). The expression heatmap also showed that a panel of ISGs are upregulated in METTL3-deficient IECs compared to WT IECs (*Figure 2b*). To map potential m6A modification sites on the mRNA transcripts of the genes that are differentially expressed in IECs, we conducted m6A RNA immuno-precipitation (RIP)-sequencing using a previously reported protocol (*Li et al., 2017*). Metagene plots showed that the m6A peak is enriched near the stop codon and the 3'-UTR region, which is consistent with the results of previous studies (*Xuan et al., 2018*; *Dominissini et al., 2012*; *Figure 2—figure supplement 1b*). We found that m6A modified one of the master regulators of IFNs, *Irf7* (*Figure 2c*). Analysis of the STRING database of predicted functional associations between proteins showed that *Irf7* lays a key role in the network of genes that were expressed differentially in METTL3-deficient IECs compared to WT

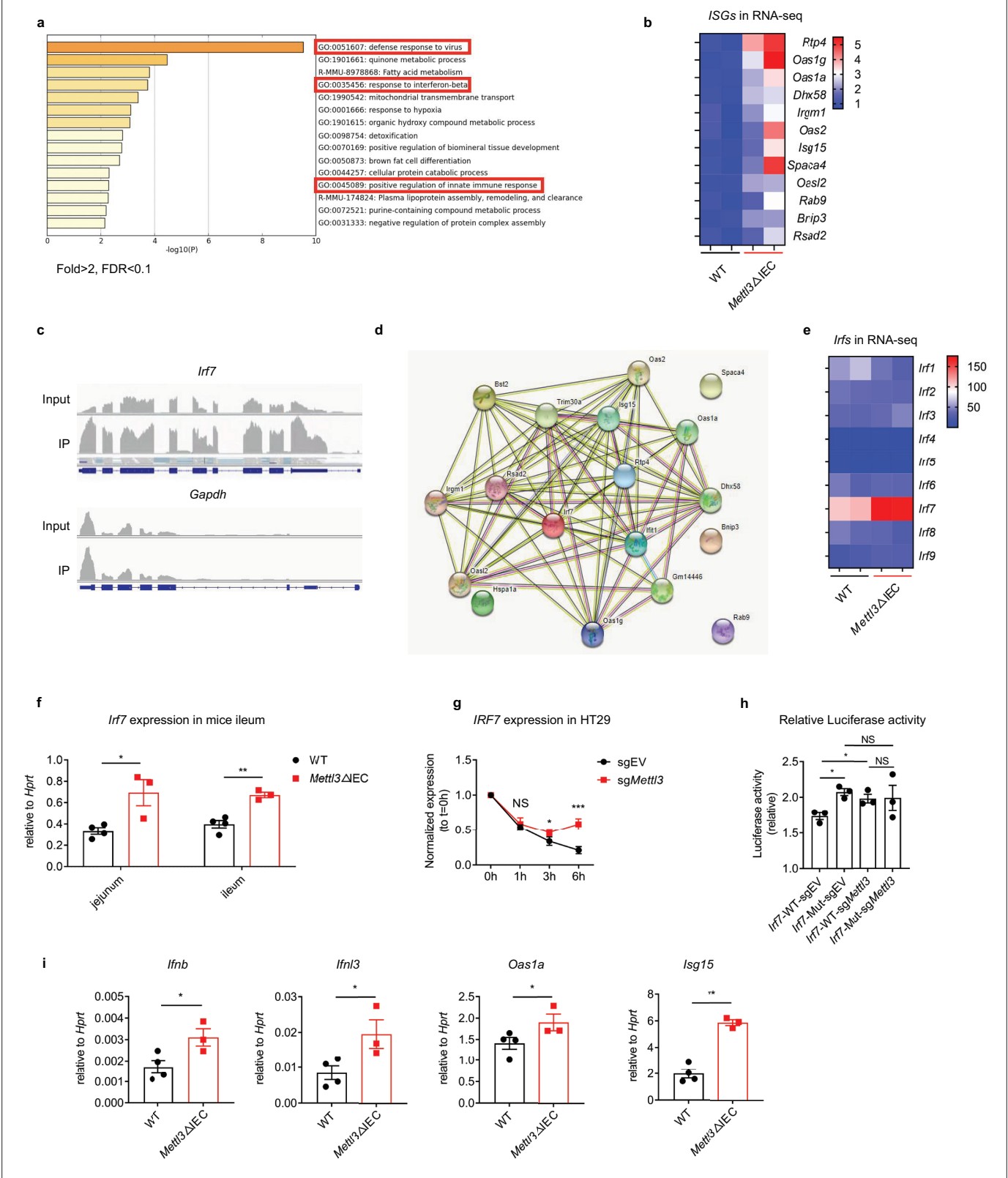

**Figure 2.** METTL3 deficiency in intestinal epithelial cells (IECs) results in decreased m6A deposition on *Irf7*, and increased interferon responses. (**a**) Gene ontology (GO) analysis of differentially expressed genes in IECs from *Mettl3*ΔIEC mice vs IECs from littermate wild-type (WT) mice. (**b**) Heat map of a subset of upregulated IFN-stimulated genes (ISGs) in IECs from *Mettl3*ΔIEC mice vs IECs from littermate WT mice, as revealed by RNA-seq (normalized data). (**c**) m6A-RIP-seq analysis of *Irf7* and (as a reference) *Gapdh* mRNA in the ileum of WT mice. (**d**) Gene regulation network of a subset of

*Figure 2 continued on next page*

*Figure 2 continued*

upregulated genes including *Irf7*. (**e**) Heat map of Interferon regulatory factor genes (*Irfs*) in IECs from *Mettl3*ΔIEC mice vs IECs from littermate WT mice, as revealed by RNA-seq (RPKM). (**f**) *Mettl3*ΔIEC mice and littermate controls were infected by EW at 8 days post birth. qPCR analysis of *Irf7* expression in the ileum and jejunum of *Mettl3*ΔIEC mice and littermate controls was carried out at 2 days post infection (dpi) (littermate WT n = 4, *Mettl3*ΔIEC mice n = 3, mean ± SEM). Statistical significance was determined by Student's t-test (*p < 0.05, **p < 0.005). (**g**) q-PCR analysis of *Irf7* mRNA in METTL3 knockdown HT-29 cells or in control cells at the indicated time points post actinomycin D treatment (n = 3, mean ± SEM). Statistical significance was determined by Student's t-test (*p < 0.05, ***p < 0.001, NS, not significant). (**h**) Relative luciferase activity of sgEV and sg*Mettl3* HEK293T cells transfected with pmirGLO-*Irf7*-3′UTR (*Irf7*-WT) or pmirGLO-*Irf7*-3′UTR containing mutated m6A modification sites (*Irf7*-MUT). The firefly luciferase activity was normalized to Renilla luciferase activity (n = 3, mean ± SEM). Statistical significance was determined by Student's t-tests between groups (*p < 0.05, NS, not significant). (**i**) *Mettl3*ΔIEC mice and littermate controls and were infected with EW at 8 days post birth. qPCR analysis of selected IFNs and ISGs in ileum tissue was carried out at 2 dpi (littermate WT, n = 4, *Mettl3*ΔIEC mice, n = 3, mean ± SEM). Statistical significance was determined by Student's t-tests between genotypes (*p < 0.05, **p < 0.005). Experiments in (**f, h and i**) were repeated three times, whereas those in (**g**) were repeated twice.

The online version of this article includes the following figure supplement(s) for figure 2:

**Figure supplement 1.** Characterization of m6A modifications on *Irf7* mRNA.

**Figure supplement 2.** METTL3 knockdown in HT-29 cells results in increased IFN response.

**Figure supplement 3.** METTL3 deficiency in MA104 cells results in increased resistance to Rhesus rotavirus infection.

**Figure supplement 4.** m6A modification on *Irf7* mRNA regulates its expression.

**Figure supplement 5.** Expression of *Irf7* and ISGs in the ileum of mice during early-life development.

---

IECs (*Figure 2d*). Of note, *Irf7* was the only *Irf* gene that was highly expressed in METTL3-deficient IECs, and prominently the highest expressed *Irf* in IECs (*Figure 2e*), indicating that *Irf7* might be one of the key genes that are modulated by m6A modifications. We also validated our results by using m6A RIP-qPCR to examine m6A modification sites in *Irf7* mRNA, based on our RIP-sequencing data from METTL3 knock-down MODE-K cells and predicted results from the database (http://rna.sysy.edu.cn) (*Figure 2—figure supplement 1a*, *Figure 2—figure supplement 1c* and *Figure 2—figure supplement 1d*). It should be noted that our m6A-RIP-seq analyses were performed at steady state. We did not identify the previously reported effects on the expression of *Ifn* genes (*Rubio et al., 2018*; *Winkler et al., 2019*), possibly due to the low expression level of *Ifn* genes in IECs at steady state.

IRF7 is a known master regulator of Type I interferon- and Type III interferon-dependent immune responses to virus infection (*Barro and Patton, 2007*; *Honda et al., 2005*; *Ciancanelli et al., 2016*). We reasoned that the loss of m6A modifications on *Irf7* mRNA is responsible for the increased IFN response and subsequent resistance to RV infection seen in METTL3-deficient mice. Thus, we first validated the regulation of *Irf7* mRNA levels by m6A in mice and in IEC cells. We found elevated levels of *Irf7* mRNA in the ileum tissue of *Mettl3*ΔIEC mice when compared to that of WT littermate control mice (*Figure 2f*). The expression of *IRF7* mRNA was also consistently higher in METTL3-knockdown HT-29 and *METTL3*-knockout rhesus monkey MA104 cells, suggesting that the regulation of *IRF7*/*Irf7* expression by m6A is likely to be conserved across species (*Figure 2—figure supplement 2*, *Figure 2—figure supplement 3*). Furthermore, genetic knockdown of METTL3 in human or mice IEC cell lines also led to increased *IRF7*/*Irf7* expression and *IFN*/*Ifn* responses (*Figure 2—figure supplement 1f*, *Figure 2—figure supplement 2*). As m6A is known to regulate the mRNA decay, we next sought to determine whether the stability of *IRF7* mRNA is regulated by m6A. We used actinomycin D to block de novo RNA synthesis in HT-29 cells, in order to assess the RNA degradation by METTL3 knockdown. The *IRF7* mRNA degraded significantly slower in METTL3-knockdown HT-29 cells than in the control cell line (*Figure 2g*, *Figure 2—figure supplement 2b*).

Luciferase reporter assays were conducted to evaluate directly the role of m6A in modulating the stability of *Irf7* mRNA. In comparison with wild-type *Irf7*-3′UTR (*Irf7*-WT) constructs, the ectopically expressed constructs harboring m6A mutant *Irf7*-3′UTR (*Irf7*-MUT) showed significantly increased luciferase activity (*Figure 2h*, *Figure 2—figure supplement 1e*). Further, mutation of m6A modification sites can directly increase the expression of *Irf7* and enhance the antiviral function of *Irf7* in MEF cells (*Figure 2—figure supplement 4a* and *Figure 2—figure supplement 4b*). These results suggest that the upregulation of *Irf7* mRNA level in *Mettl3*ΔIEC mice is caused by the loss of mRNA decay mediated by m6A modification. To evaluate the potential influence of m6A on the transcriptional targets of *Irf7*, we also measured the expression of IFNs and ISGs in rotavirus-infected ileum tissue from *Mettl3*ΔIEC mice and littermate WT mice. We found that the transcriptional targets of *Irf7*, were all upregulated in *Mettl3*ΔIEC mice (*Figure 2i*). Furthermore, we found that the mRNAs of *Irf7* and

its transcriptional-target ISGs increased in the ileum of the mice from 1 to 4 weeks post birth, with a dramatic upregulation from 2 weeks to 3 weeks post birth (*Figure 2—figure supplement 5*), which was concomitant with the decrease in the frequency of global m6A modifications (*Figure 1a and b* and *Figure 1—figure supplement 1a*). These results demonstrated that METTL3 deficiency in IECs results in decreased m6A deposition on *Irf7*, and in increased interferon response.

## IRF7 deficiency attenuated the increased interferon response and resistance to RV infection in *Mettl3*ΔIEC mice

To determine whether *Irf7* plays a key role in the RV-infection-resistant phenotype of the IECs of METTL3-deficient mice, we crossed *Irf7*⁻/⁻ mice to *Mettl3*ΔIEC mice. Following RV oral gavage, the expression levels of IFNs and ISGs in ileum tissue from *Irf7*⁻/⁻*Mettl3*ΔIEC mice were significantly lower than those in the corresponding tissues from *Mettl3*ΔIEC mice at 2 dpi (*Figure 3a–c*). Unlike the increased expression of IFNs and ISGs in *Mettl3*ΔIEC mice vs littermate WT controls caused by IRF7 deficiency, the expression levels of IFNs and ISGs in ileum from *Irf7*⁻/⁻*Mettl3*ΔIEC mice were not significantly different from those in ileum from *Irf7*⁻/⁻ mice (*Figure 3a–c*), suggesting that *Irf7* mediates the increased expression of IFNs and ISGs in *Mettl3*ΔIEC mice.

Moreover, the *Irf7*⁻/⁻*Mettl3*ΔIEC mice showed significantly higher viral loads in ileum tissue and higher fecal shedding of RV *Mettl3*ΔIEC (*Figure 3d–f*). Similarly, unlike the much lower fecal shedding of virus from *Mettl3*ΔIEC mice vs littermate WT controls that resulted from IRF7 deficiency, the fecal viral shedding from *Irf7*⁻/⁻*Mettl3*ΔIEC mice was not significantly different from that from *Irf7*⁻/⁻ mice (*Figure 3d*), suggesting that *Irf7* mediates the rotavirus-infection-resistant phenotype measured by fecal viral shedding in *Mettl3*ΔIEC mice. Notably, the difference in the expression of viral proteins in the ileum was much lower when comparing *Irf7*⁻/⁻*Mettl3*ΔIEC mice with *Irf7*⁻/⁻ mice (9.7-fold lower for NSP2 and 9.3-fold lower for VP7) than when comparing *Mettl3*ΔIEC mice with littermate mice (267.1-fold lower for NSP2 and 283.4-fold lower for VP7) (*Figure 3e*). This suggests that other pathways (e.g. m6A modifications in RV RNA), besides that mediated by *Irf7*, may also play roles in the rotavirus-infection-resistant phenotype of IECs from *Mettl3*ΔIEC mice. In summary, IRF7 is an important mediator of the increased expression of IFN and ISGs in *Mettl3*ΔIEC mice, and genetic deletion of IRF7 restored the resistant-to-RV-infection phenotype of *Mettl3*ΔIEC mice.

## Rotavirus suppresses ALKBH5 expression through NSP1 to evade immune defense

We next sought to determine how RV regulates m6A modifications in IECs. We first measured whether RV infection regulates the m6A-related writer and eraser proteins in the intestine. The protein levels of the methyltransferases METTL3 and METTL14 and the demethylase FTO were not affected by RV infection in ileum tissue (*Figure 4a and b*). By contrast, the protein level of demethylase ALKBH5 was significantly downregulated by RV infection in the ileum (*Figure 4a and b*). To determine whether ALKBH5 plays a role in anti-RV infection, we generated an IEC-specific deletion of ALKBH5 in mice (*Alkbh5*^fl/fl *Vil1*^Cre, *Alkbh5*ΔIEC). The depletion of ALKBH5 in IECs did not affect the anti-RV immune response (*Figure 4c*), viral shedding in the feces (*Figure 4d*), or viral protein expression in the ileum (*Figure 4e*), probably because of the suppressed expression of ALKBH5 in the ileum tissue of WT mice infected by RV.

We next asked whether there is redundancy between the m6A erasers ALKBH5 and FTO. We first checked the expression of *Alkbh5/ALKBH5* and *Fto/FTO* in our RNA-seq data from mouse IECs, and also in previously reported RNA-seq data from human intestinal enteroids (*Saxena et al., 2017*). We found that *Alkbh5/ALKBH5* expression is much higher than *Fto/FTO* expression in intestine (*Figure 4—figure supplement 1a*). We then carried out qPCR analysis in the mouse epithelial cell line MODE-K, which showed the similar results (*Figure 4—figure supplement 1a*). Furthermore, we next knocked down METTL3, ALKBH5 and FTO in MODE-K cells. Using a dot blot assay, we found that ALKBH5 rather than FTO is the dominant m6A eraser in IECs (*Figure 4—figure supplement 1*). Furthermore, overexpression of recombinant ALKBH5 increased the IFN/ISGs response and inhibited Rhesus RV infection of mouse IEC MODE-K cells (*Figure 4—figure supplement 2*), suggesting that re-expression of ALKBH5 might overcome the immune evasion of RV.

Non-structural protein 1 (NSP1) is a well-established RV-encoded innate immune antagonist that has been shown to degrade IRF3 and β-Trcp (*Barro and Patton, 2005*; *Ding et al., 2016*). To test

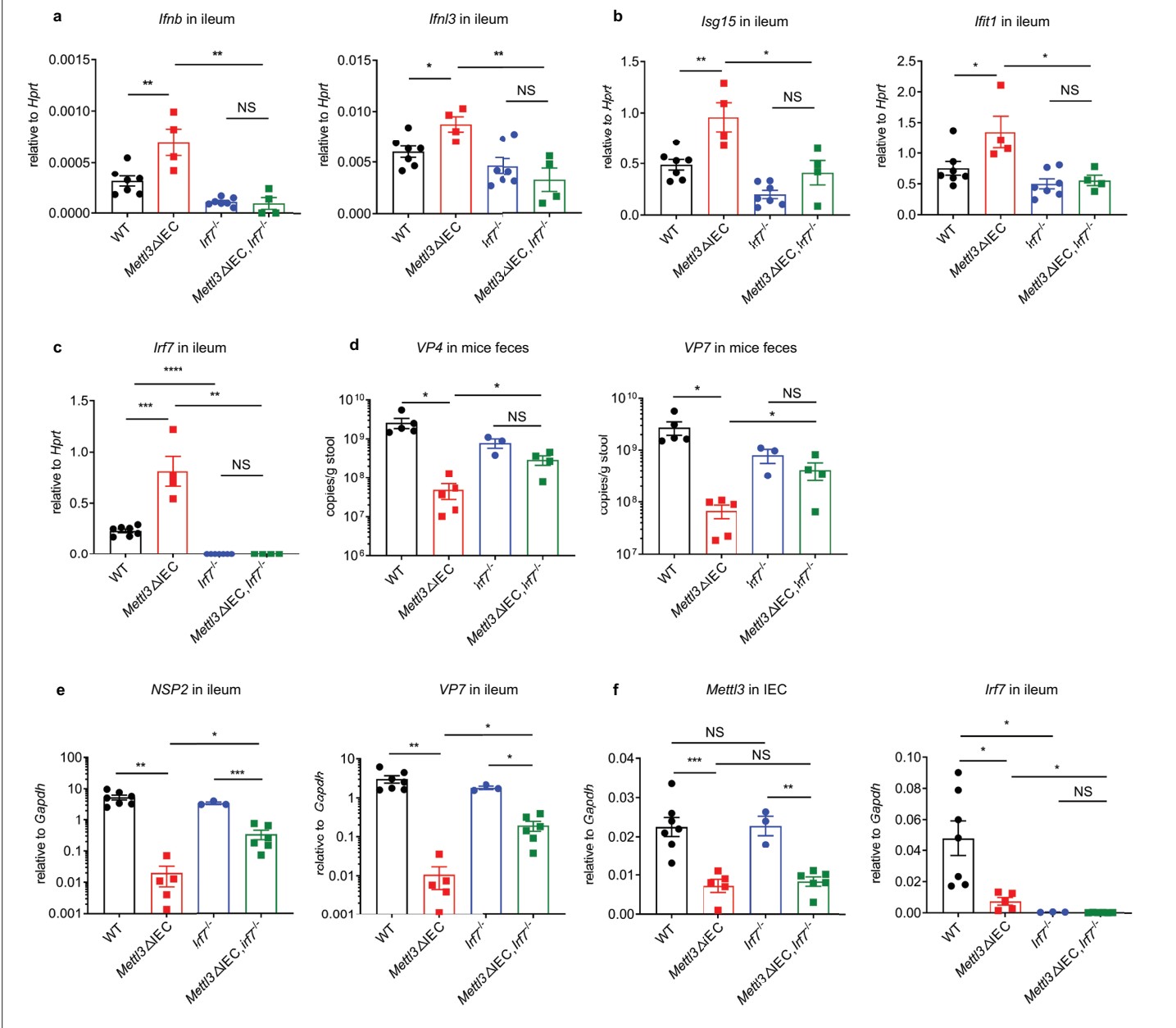

**Figure 3.** IRF7 deficiency attenuated the increased interferon response and resistance to rotavirus infection in *Mettl3*ΔIEC mice. (**a–c**) The wild-type (WT) control mice, *Mettl3*ΔIEC mice, *Irf7*-/- mice and *Mettl3*ΔIEC *Irf7*-/- mice are all littermates. They were infected by RV EW at 8 days post birth. The expression of selected interferon (IFN) genes (**a**), IFN-stimulated genes (ISGs) (**b**), or *Irf7* (**c**) in ileum from indicated groups of mice was analyzed by qPCR at 2 dpi (littermate WT n = 7, *Mettl3*ΔIEC n = 5, *Irf7*-/- n = 3, *Mettl3*ΔIEC *Irf7*-/- n = 6, mean ± SEM). Statistical significance was determined by Student's t-tests between genotypes (*p < 0.05, **p < 0.005, ***p < 0.001, ****p < 0.0001, NS, not significant). (**d**) Fecal rotaviral shedding by the indicated groups of mice was analyzed by qPCR at 4 dpi (littermate WT n = 5, *Mettl3*ΔIEC n = 5, *Irf7*-/- n = 3, *Mettl3*ΔIEC *Irf7*-/- n = 4, mean ± SEM). Statistical significance was determined by Student's t-tests between genotypes (*p < 0.05, NS, not significant). (**e–f**) Expression of RV proteins (**e**) or *Mettl3* and *Irf7* (**f**) in ileum tissue or IECs from the indicated groups of mice was analyzed by qPCR at 4 dpi (littermate WT n = 7, *Mettl3*ΔIEC n = 5, *Irf7*-/- n = 3, *Mettl3*ΔIEC *Irf7*-/- n = 6, mean ± SEM). Statistical significance was determined by Student's t-tests between genotypes (*p < 0.05, **p < 0.005, ***p < 0.001, NS, not significant). Experiments in (**a–f**) were repeated twice.

the potential role of NSP1 in AlKBH5 inhibition, we used the recently developed reverse genetics system and rescued a recombinant WT RV SA11 strain and a mutant virus that did not express NSP1 (NSP1-null) (*Kanai et al., 2017*). We infected HEK293 cells with WT and NSP1-null RVs, and found that only WT RV reduced the ALKBH5 protein levels (*Figure 4f*), suggesting that the downregulation of

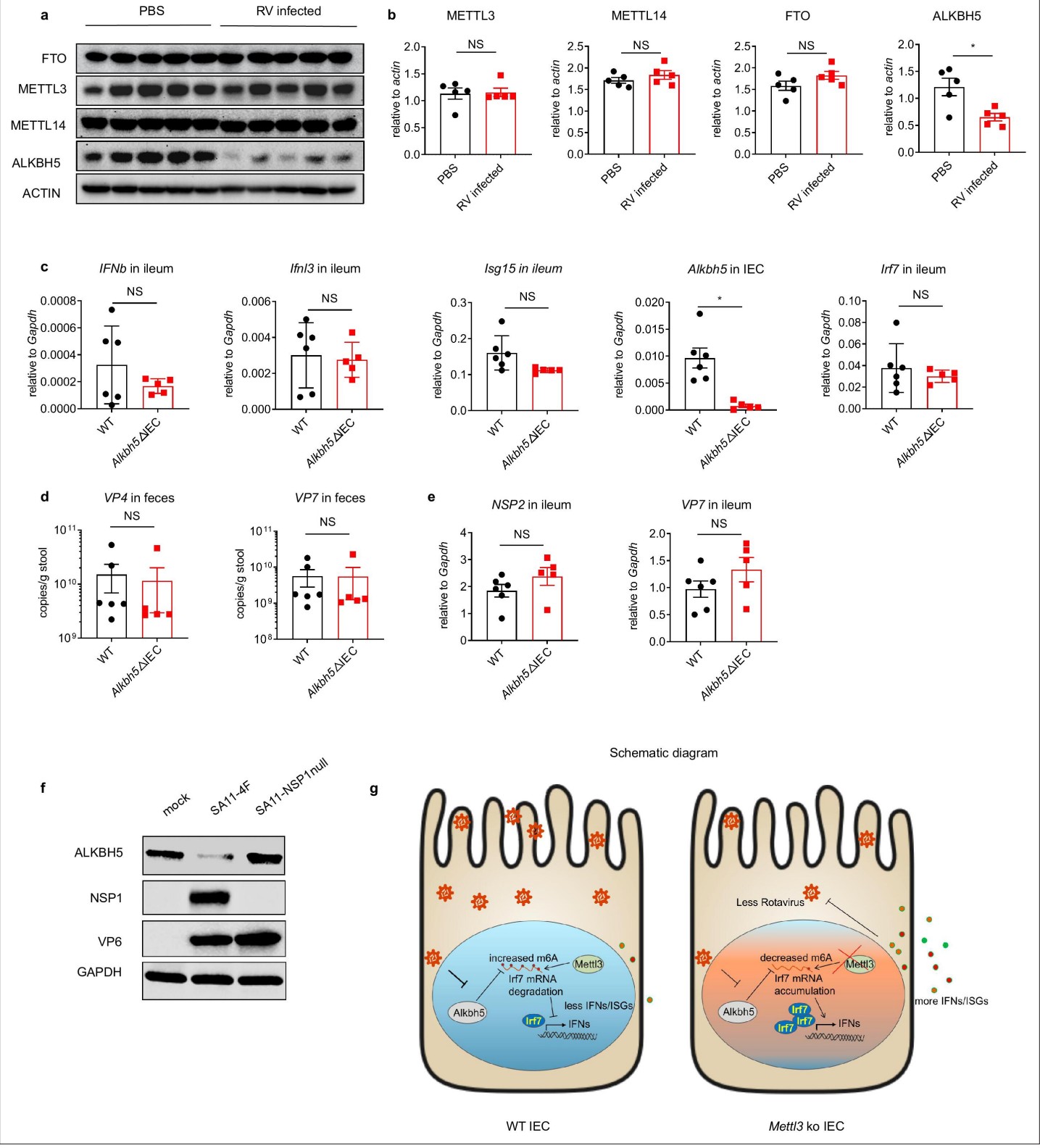

**Figure 4.** Rotavirus suppresses ALKBH5 expression through NSP1 to evade immune defense. (**a**) WT mice were infected by RV EW at 8 days post birth. Immunoblotting with antibodies targeting ALKBH5, FTO, METTL14 and METTL3 in ileum tissue from mice infected with RV EW or treated with PBS was carried out at 2 days post infection (dpi). (**b**) Quantitative analysis of the immunoblot shown in (**a**) (mean ± SEM). Statistical significance was determined by Student's t-test (*p < 0.05, NS, not significant). (**c–e**) *Alkbh5*ΔIEC mice and littermate controls were infected by RV EW at 8 days post birth. The expression of the indicated genes in ileum tissue or IECs (**c**), viral shedding in feces (**d**), and the expression of viral proteins in ileum tissues

*Figure 4 continued on next page*

*Figure 4 continued*

(**e**), were analyzed for *Alkbh5*ΔIEC mice or littermate controls at 4 dpi (littermate WT n = 6, *Alkbh5*ΔIEC n = 5, mean ± SEM). Statistical significance was determined by Student's t-tests between genotypes (*p < 0.05, NS, not significant). (**f**) Immunoblotting with antibodies targeting ALKBH5, NSP1, VP6 and GAPDH in HEK293 cells infected by SA11-4F and SA11-NSP1null (MOI = 1) for 24 hr. (**g**) Graphical abstract illustrating the functions and molecular mechanisms of m6A modification on *Irf7* in anti-RV infection. Experiments in (**a–e**) were repeated three time, whereas those in (**f**) were repeated twice.

The online version of this article includes the following figure supplement(s) for figure 4:

**Figure supplement 1.** A non-redundant role of *Alkbh5* in regulating m6A modification in intestinal epithelial cells (IECs).

**Figure supplement 2.** ALKBH5 overexpression in MODE-K cells results in enhanced IFN response and increased resistance to Rhesus rotavirus infection.

**Figure supplement 3.** m6A-RIP-qPCR analysis of the predicted m6A sites on Rotavirus RNA.

**Figure supplement 4.** METTL3 deficiency leads to aberrant double-stranded RNA (dsRNA) formation in isolated IECs.

ALKBH5 expression by RV is NSP1-dependent. These results suggest that RV might evade the antiviral immune response via downregulation of ALKBH5 expression by NSP1.

## Discussion

Previous studies have reported that m6A modifications on mRNA in mice embryonic fibroblasts or normal human dermal fibroblasts negatively regulate the IFN response by accelerating the mRNA degradation of type I IFNs (*Rubio et al., 2018*; *Winkler et al., 2019*). However, these studies were mainly conducted in vitro, leaving the relationship between m6A and the IFN pathway in vivo as unexplored territory. As type I and type III IFNs play a crucial role in the antiviral immune response in the gastrointestinal tract (*Pott et al., 2011*), we used an RV infection model in which IECs are specifically infected, as well as conditional knockout mice with IECs-specific depletion of m6A writer METTL3 or m6A eraser ALKBH5, to study the role of m6A modification in regulating IFN response towards rotavirus in IECs. Using RNA-seq and m6A-seq techniques, we identified IRF7, a key transcription factor upstream of IFNs and ISGs, as one of the targets that are modified by m6A during RV infection and as an essential mediator of the elevated anti-rotavirus immune response that results from METTL3 deficiency. These results identify *Irf7* as an important m6A target, and characterize, for the first time, the regulation of the IFN response during RV infection in the intestine, thereby providing a better understanding of how m6A modifications on mRNA regulate antiviral innate immune responses.

In addition to regulating target genes that are involved in innate immune pathways directly, m6A modification is also known to affect viral gene expression, viral replication, and the generation of progeny virions (*Roundtree et al., 2017*; *Brocard et al., 2017*). Some viral RNA genomes are modified by m6A, such as those of simian virus 40 (*Tsai et al., 2018*; *Lavi and Shatkin, 1975*), influenza A virus (*Krug et al., 1976*), adenovirus (*Sommer et al., 1976*), avian sarcoma virus (*Dimock and Stoltzfus, 1977*), Rous Sarcoma virus (*Kane and Beemon, 1985*), hepatitis C virus (*Gokhale et al., 2016*) and Zika virus (*Lichinchi et al., 2016*). Our m6A-RIP-qPCR showed that Rotavirus RNA also has m6A modification (*Figure 4—figure supplement 3*). Of note, genetic deletion of METTL3 in the monkey kidney MA104 cell line, which has limited IFN responses (*Sánchez-Tacuba et al., 2020*), also led to reduced RV replication (*Figure 2—figure supplement 3*). Further, IRF7 deficiency did not fully restore the suppression of rotaviral infection in *Mettl3*ΔIEC mice (*Figure 3e*). These results suggest that other pathways (e.g. m6A modifications in RV RNA), in addition to that mediated by *Irf7*, may also contribute to the resistant-to-rotavirus-infection phenotype of IECs from *Mettl3*ΔIEC mice. The detailed mechanisms warrant further investigation in the future.

Intriguingly, m6A modifications have been shown to maintain the genomic stability of mice embryonic stem cells by promoting the degradation of endogenous retrovirus (ERV) mRNA, or by regulating ERV heterochromatin and inhibiting its transcription (*Xu et al., 2021*; *Chelmicki et al., 2021*). The absence of m6A results in the formation of abnormal endogenous double-stranded RNA (dsRNA), which causes an aberrant immune response and necrosis in the hematopoietic system (*Gao et al., 2020*). In the intestine, through immunostaining with the J2 antibody, we detected increased dsRNA levels in the IECs of *Mettl3*ΔIEC mice when compared to those of littermate WT mice (*Figure 4— figure supplement 4*). Consistent with this, the RNA-seq data showed that the expression of a set of ISGs, including *Irf7*, was significantly upregulated in *Mettl3* KO IECs. Our data suggest a dual

activation model in the steady state: in the absence of METTL3, an increase in the level of dsRNA will induce IFN responses, and the increased stability of *Irf7* mRNA, which encodes a key transcription factor in IFN and ISG expression, will amplify this process. Furthermore, m6A modification of viral RNA genomes affects the activation of innate sensor-mediated signaling (*Chen et al., 2021*; *Kim et al., 2020*). Decreased m6A modification on RV genomes may activate innate sensors directly and could induce a greater IFN response, which will also be amplified by the increased stability of *Irf7* mRNA.

Although m6A is involved in many important biological processes, the regulation of m6A modifications remains poorly understood. Here, we found that RV infection downregulates the expression of the m6A eraser ALKBH5, thereby inducing m6A modifications, in an NSP1-dependent manner. The precise mechanism remains to be elucidated. As a result, ALKBH5 deficiency in IECs results in normal susceptibility to RV infection. In addition, global m6A modification of mRNA transcripts declines in the intestine as newborn mice age, with a significant drop from 2 to 3 weeks post birth. This implies that there is a fall in RV infectivity in adult mice vs neonatal mice. The dual regulation of m6A levels during RV infection and development provides new insights into mechanisms available to both the RV and the host to regulate m6A modification in order to achieve either immune evasion or immune surveillance, respectively.

In conclusion, our work sheds light on a novel role of m6A modifications in RV infection in vivo, and reveals a tissue-specific regulation of m6A during RV infection and mouse neonatal development (*Figure 4g*). Future studies on the tissue-specific regulation of m6A modification by viral infections in other tissues and organs (e.g. lung, liver) will be of interest.

## Materials and methods
### Mice
METTL3 conditional knockout mice were generated by inserting two loxp sites into the intron after the first exon and the intron before the last exon of *Mettl3* using the CRISPR/cas9-based genome-editing system as previously described (*Li et al., 2017*). ALKBH5 conditional knockout mice were generated by inserting two loxp sites into the introns flanking the first exon of *Alkbh5* using the CRISPR/cas9-based genome-editing system as previously described (*Zhou, 2021*). The genotypes of *Mettl3*^fl/fl mice, *Alkbh5*^fl/fl mice, *Vil1*^Cre mice (The Jackson Laboratory, Stock No: 021504), and *Irf7*^-/- mice (RIKEN BRC, RBRC01420) were confirmed by PCR using the primers listed below:

*Mettl3*^fl/fl mice

> *Mettl3*-L1+: CCCAACAGAGAAACGGTGAG
> *Mettl3*-L2-: GGGTTCAACTGTCCAGCATC

*Vil1*^Cre mice

> *Vil1*^Cre-182/150 F: GCCTTCTCCTCTAGGCTCGT
> *Vil1*^Cre-182-R: TATAGGGCAGAGCTGGAGGA
> *Vil1*^Cre-150-R: AGGCAAATTTTGGTGTACGG

*Irf7*^-/- mice

> RBRC01420-*Irf7*-WT-F: GTGGTACCCAGTCCTGCCCTCTTTATAATCT
> RBRC01420-*Irf7*-Mut-F: TCGTGCTTTACGGTATCGCCGCTCCCGATTC
> RBRC01420-*Irf7*-R: AGTAGATCCAAGCTCCCGGCTAAGTTCGTAC

*Alkbh5*^fl/fl mice

> *Alkbh5*-L1+: GCACAGTGGAGCACATCATG
> *Alkbh5*-L2-: CAGAGGGCAAGCAACCACAC

The sex-, age-, and background-matched littermates of the knockout or conditional knockout mice were used as the controls in the present study. All mice were on the C57BL/6 background. Mice were

maintained in specific-pathogen-free (SPF) conditions under a strict 12 hr light cycle (lights on at 08:00 and off at 20:00). All animal studies were performed according to protocols approved by the Ethics Committee at the University of Science and Technology of China (USTCACUC202101016).

## Cell culture

The MA104 cell line was obtained from the Cell Resource Center, Peking Union Medical College (which is the headquarters of the National Infrastructure of Cell Line Resource). HEK293T (ATCC CRL-3216) and HT-29 (ATCC HTB-38D) cells were obtained from the American Type Culture Collection (ATCC). MODE-K cells were obtained from Shanghai HonSun Biological Technology Co., Ltd.

The identities of these cell lines were authenticatedby short tandem repeat (STR) profiling (FBI, CODIS). All of these cells were cultured in Dulbecco's modified Eagle's medium (DMEM) (Hyclone) supplemented with 10% fetal bovine serum (FBS) (Clark). All cells were cultured at 37 °C in 5% $CO_2$. All cells were tested to eliminate the possibility of mycoplasma contamination.

## Poly(I:C) transfection

Poly(I:C) (OKA, A55994) was transfected at a concentration of 2 µg/ml using Lipofectamine 3,000 (Invitrogen) according to the manufacturer's protocol. Cells were analyzed at the indicated times after birth or infection.

## Plasmids and SgRNAs

pSIN-m*Alkbh5*-3xflag and pLVX-m*Alkbh5*-puro were constructed by cloning mouse *Alkbh5* CDS into pSIN-3xflag and pLVX-puro plasmid.
pLVX-m*Irf7*-wt-puro was constructed by cloning mouse *Irf7* CDS and 3'UTR into pLVX-puro.
pLVX-m*Irf7*-Mut-puro was constructed by cloning mutated mouse *Irf7* CDS and 3'UTR into pLVX-puro. (The potential m6A modification sites (1400 A, 2213 A, 2693 A, 2927 A, 3446 A, 3475 A, 3491 A, 3590 A and 3620 A of mice *Irf7*(NC_000073)) were predicted on the SRAMP website. The A to G mutation in these sites did not change the encoded amino acids).

All gene silencing was done using a CRISPR–cas9 system, with a lentiCRISPR v2 plasmid (Addgene no. 52961). The following sgRNAs were cloned downstream of the U6 promoter:

Human, rhesus and mouse *METTL3*/*Mettl3*: GGACACGTGGAGCTCTATCC;
Mouse *Alkbh5*: CCTCGTAGTCGCTGCGCTCG;
Mouse *Fto*: GAAGCGCGTCCAGACCGCGG;

Lentiviruses were generated by co-transfection of lentiCRISPR v2 constructs and packaging plasmids (psPAX2, Addgene no. 12,260 and pMD2.G, Addgene no. 12259), using PEI DNA transfection reagent (Shanghai maokang biotechnology), into HEK293T cells, according to the manufacturer's instructions. At 48 hr post transfection, supernatants were collected and filtered through a 0.22 µm polyvinylidene fluoride filter (Millex). To induce gene silencing, cells were transduced with lentivirus expressing sgRNA and were puromycin selected (2 µg/ml) for 4–5 days. The depletion of target proteins was confirmed by immunoblot analysis.

## Virus infections

Rhesus and simian RV strains, including RRV (Rhesus), SA11-4F (simian), SA11-NSP1null (simian), were propagated in MA104 cells as previously described (*Ding et al., 2016*). Viruses were activated by trypsin (5 µg/ml) at 37 °C for 30 min prior to infection. Cells were washed with PBS three times and incubated with RV at different multiplicities of infection (MOIs) at 37 °C for 1 hr. After removal of RV inoculum, cells were washed with PBS, cultured in serum-free medium (SFM) and harvested for qPCR and western blot analysis at the indicated time points.

EW stock virus was prepared by infecting 5-day-old C57BL/6 J mice, and harvesting crude centrifugation-clarified intestinal homogenate as previously described (*Ding et al., 2018*).

For all rotavirus infections, except where indicated elsewhere, 8-day-old wild-type mice or genetically deficient mice were orally inoculated by gavage with RV EW virus, as previously described (*Ding et al., 2018*). Mice were sacrificed, and stool and small intestinal tissue were collected, at the indicated time points post infection. Viral loads in the intestinal tissues and feces were detected by RT–qPCR.

## Quantitative analysis of m6A level

The m6A analysis by LC-MS/MS was performed by Metware Biotechnology Co., Ltd (Wuhan, China). In brief, 1 µg purified RNA was digested into single nucleosides with 1 U nuclease P1 (Takara) and 1 U rSAP (Takara) and incubated at 37 °C. The digested RNA was injected into a LC-MS/MS, which was used to perform ultra-performance liquid chromatography with a C18 column. The positive ion multiple reaction-monitoring (MRM) mode was adopted to detect m6A abundance. m6A levels were calculated from a standard curve, which was generated from pure nucleoside standards.

## RT-qPCR

For cells and tissues, total RNA was extracted with TRNzol Universal reagent (Tiangen) in accordance with the manufacturer's instructions. Real-time PCR was performed using SYBR Premix Ex Taq II (Tli RNaseH Plus) (Takara) and complementary DNA was synthesized with a PrimeScript RT reagent Kit with gDNA Eraser (Takara). The target genes were normalized to the housekeeping gene (*Gapdh* or *HPRT*) shown as $2^{-\Delta Ct}$. The primers were as follows:

Primers to detect mouse genes

> *Mettl3*-F: ATTGAGAGACTGTCCCCTGG
> *Mettl3*-R: AGCTTTGTAAGGAAGTGCGT
> *Mettl14*-F: AGACGCCTTCATCTCTTTGG
> *Mettl14*-R: AGCCTCTCGATTT CCTCTGT
> *Fto*-F: CTGAGGAAGGAGTGGCATG
> *Fto*-R: TCTCCACCTAAGACTTGTGC
> *Alkbh5*-F: ACAAGATTAGATGCACCGCG
> *Alkbh5*-R: TGTCCATTTCCAGGATCCGG
> *Wtap*-F: GTTATGGCACGGGATGAGTT
> *Wtap*-R: ATCTCCTGCTCTTTGGTTGC
> *Gapdh*-F: TGAGGCCGGTGCTGAGTATGTCG
> *Gapdh*-R: CCACAGTCTTCTGGGTGGCAGTG
> *Hprt*-F: ACCTCTCGAAGTGTTGGATACAGG
> *Hprt*-R: CTTGCGCTCATCTTAGGCTTTG
> *Irf7*-F: ATGCACAGATCTTCAAGGCCTGGGC
> *Irf7*-R: GTGCTGTGGAGTGCACAGCGGAAGT
> *Isg15*-F: GGTGTCCGTGACTAACTCCAT
> *Isg15*-R: TGGAAAGGGTAAGACCGTCCT
> *Oas1a*-F: GCCTGATCCCAGAATCTATGC
> *Oas1a*-R: GAGCAACTCTAGGGCGTACTG
> *Ifnb*-F: TCCGAGCAGAGATCTTCAGGAA
> *Ifnb*-R: TGCAACCACCACTCATTCTGAG
> *Ifnl3*-F: AGCTGCAGGCCTTCAAAAAG
> *Ifnl3*-R: TGGGAGTGAATGTGGCTCAG
> *Ifit1*-F: GAACCCATTGGGGATGCACAACCT
> *Ifit1*-R: CTTGTCCAGGTAGATCTGGGCTTCT

Primers to detect human genes

> *GAPDH*-F: ATGACATCAAGAAGGTGGTG
> *GAPDH*-R: CATACCAGGAAATGAGCTTG
> *IRF7*-F: CGAGACGAAACTTCCCGTCC
> *IRF7*-R: GCTGATCTCTCCAAGGAGCC
> *IFNL3*-F: TAAGAGGGCCAAAGATGCCTT
> *IFNL3*-R: CTGGTCCAAGACATCCCCC
> *CXCL10*-F: TGGCATTCAAGGAGTACCTC
> *CXCL10*-R: TTGTAGCAATGATCTCAACACG
> *IFIT1*-F: CAACCATGAGTACAAATGGTG

*IFIT1*-R: CTCACATTTGCTTGGTTGTC

## Primers to detect rhesus genes

Rhesus-*GAPDH*-F: ATGACATCAAGAAGGTGGTG
Rhesus-*GAPDH*-R: CATACCAGGAAATGAGCTTG
Rhesus-*IFIT1*-F: CAACCATGAGTACAAATGGTG
Rhesus-*IFIT1*-R: CTCACACTTGCTTGGTTGTC
Rhesus-*IRF7*-F: GTTCGGAGAGTGGCTCCTTG
Rhesus-*IRF7*-R: TCACCTCCTCTGCTGCTAGG
Rhesus-*IFNL1*-F: ACTCATACGGGACCTGACAT
Rhesus-*IFNL1*-R: GGATTCGGGGTGGGTTGAC
Rhesus-*IFNb*-F: GAGGAAATTAAGCAGCCGCAG
Rhesus-*IFNb*-R: ATTAGCAAGGAAGTTCTCCACA

## Primers to detect virus genes

Rotavirus EW-NSP2-F: GAGAATGTTCAAGACGTACTCCA
Rotavirus EW-NSP2-R: CTGTCATGGTGGTTTCAATTTC
Rotavirus EW-VP4-F: TGGCAAAGTCAATGGCAACG
Rotavirus EW-VP4-R: CCGAGACACTGAGGAAGCTG
Rotavirus EW-VP7-F: TCAACCGGAGACATTTCTGA
Rotavirus EW-VP7-R: TTGCGATAACGTGTCTTTCC
RRV VP7-F: ACGGCAACATTTGAAGAAGTC
RRV VP7-R: TGCAAGTAGCAGTTGTAACATC
RRV NSP2-F: GAGAATCATCAGGACGTGCTT
RRV NSP2-R: CGGTGGCAGTTGTTTCAAT
RRV NSP5-F: CTGCTTCAAACGACCCACTCAC
RRV NSP5-R: TGAATCCATAGACACGCC

## m6A dot blot assay

Total RNA was isolated from mice ileum using TRNzol Universal Reagent (Tiangen, Lot#U8825) according to the manufacturer's instructions. RNA samples were quantified using UV spectrophotometry and denatured at 95 °C for 3 min. The m6A-dot-blot was performed as described previously (*Shen et al., 2016*). In brief, the primary rabbit anti-m6A antibody (1:5000, Synaptic System, #202003) or the primary rabbit anti-m6A antibody (1:1000, Sigma-Aldrich, ABE572-I-100UG) was applied to an Amersham Hybond-N+ membrane (GE Healthcare, USA) containing the RNA samples. Dot blots were visualized by the imaging system after incubation with secondary antibody HRP-conjugated goat anti-rabbit IgG (Beyotime, A0208).

## Western blot

Briefly, cells and tissue were lysed with RIPA buffer (Beyotime Biotechnology) supplemented with PMSF (Beyotime Biotechnology) and protease inhibitor cocktail (Roche). METTL3 (Abcam, ab195352, 1:2000), METTL14 (Sigma, HPA038002, 1:2000), ALKBH5 (Sigma, HPA007196, 1:2000), FTO (Abcam, ab92821), NSP1 and VP6 (gifts from Harry B. Greenberg's lab), GAPDH (Proteintech), TUBULIN (Proteintech), beta-ACTIN (Proteintech), Phospho-IRF-7 (Ser437/438) (D6M2I) (CST), Phospho-TBK1/NAK (Ser172) (D52C2) (CST), TBK1/NAK (D1B4) (CST), and IRF-7 (D8V1J) (CST) antibodies were used in accordance with the manufacturer's instructions. After incubation with the primary antibody overnight, the blotted PVDF membranes (Immobilon, IPVH00010) were incubated with goat anti-rabbit IgG-HRP (Beyotime, A0208) or goat anti-mouse IgG-HRP (Beyotime, A0216) and exposed with the BIO-RAD ChemiDocTM Imaging System for the correct exposure period.

## RNA degradation assay

The stability of the targeted mRNA was assessed as previously described (*Li et al., 2017*). In brief, METTL3 knockdown HT-29 cells and control cells were plated in a 24-well plate. Actinomycin-D (MCE,

HY17559) was added to a final concentration of 5 μM, and cells were harvested at the indicated time points after actinomycin-D treatment. The RNA samples were processed and qPCR was used to measure the number of mRNA transcripts; all data were normalized to those at the t = 0 time point.

## Dual-luciferase assay

The pmirGLO (Firefly luciferase, hRluc) vector of the Dual-luciferase Reporter assay system (Promega, E1910) was used to determine the function of m6A modification within the 3'UTR of *Irf7* transcripts. The potential m6A modification sites (3446 A, 3475 A, 3491 A, 3590 A, and 3620 A of mice *Irf7*(NC_000073)) were predicted on the SRAMP website. The mutants harbor A to G mutations at these sites. The assay was performed according to the manufacture's instructions for the Promega Dual-Luciferase Reporter Assay System (E1910). Briefly, 300 ng of pmirGLO vector containing *Irf7*-3'UTR or m6A-mutant *Irf7*-3'UTR were transfected into HEK293T cells in triplicate wells. The relative luciferase activity was accessed 36 hr post transfection.

## Isolation of IECs in the intestine

Small intestines were excised and flushed thoroughly three times with PBS. They were turned inside out and cut into ~1 cm sections, then transferred into RPMI with 2 mM EDTA, and shaken for 15 min at 37 °C. Supernatants were collected through a 100-mm cell strainer to obtain single-cell suspensions. Cells were collected as an IEC fraction, which contains both epithelial cells (~90%) and lymphocytes (IEL, ~ 10%). The single-cell suspension was used for further analysis.

## RNA-Seq

IECs from *Mettl3*ΔIEC mice and from wild-type littermate control mice were isolated as described in the previous section. Total RNAs were extracted with TRNzol universal RNA Reagen kits. Berrygenomics (Beijing, China) processed the total RNA and constructed the mRNA libraries, and subjected them to standard Illumina sequencing on a Novaseq 6,000 system, obtaining >40 million pair-end 150 reads for each sample. Raw RNA-sequencing reads were aligned to the mouse genome (mm10, GRCm38) with STAR (v2.5.3a). Gene expression levels were determined and differential analysis was performed with edgeR(v3.29.2). Genes were considered significantly differentially expressed if they showed ≥1.5-fold change and a false discovery rate (FDR) <0.05. Gene set analysis was performed and enriched pathways were obtained through online bioinformatics tools (metascape) and GSEA (v4.0.3). Pathway plots were gene-rated with R package 'ggplot2' (*Li et al., 2017*).

## m6A RNA-IP-qPCR and m6A RNA-IP-Seq

m6A RNA-IP-Seq was carried out according to a previously published protocol (*Li et al., 2017*). In brief, total cellular RNA extracted from WT C57 mice IEC was fragmented by ZnCl2 followed by ethanol precipitation. Fragmented RNA was incubated with an anti-m6A antibody (Sigma Aldrich ABE572-I) or IgG IP Grade Rabbit polyclonal antibody (Abcam, lot: 934197). The eluted RNA and input were subjected to high-throughput sequencing using standard protocols (Illumina, San Diego, CA, USA) or processed as described in the 'RT-qPCR' section, except that the data were normalized to the input samples. The m6A RIP-Seq data were analyzed as described previously (*Li et al., 2017*).

> RIP-*Ptpn4*-F: CCTCCCATCCCGGTCTCCACC
> RIP-*Ptpn4*-R: GGCTGCCCATCTTCAGGGGT
> RIP-*Rps14*-F: ACCTGGAGCCCAGTCAGCCC
> RIP-*Rps14*-R: CACAGACGGCGACCACGACG
> RIP-*Tlr3*-F: TGCTCAGGAGGGTGGCCCTT
> RIP-*Tlr3*-R: CGGGGTTTGCGCGTTTCCAG
> m6A-*Irf7*-F1: GACAGCAGCAGTCTCGGCTT
> m6A-*Irf7*-R1: ACCCAGGTCCATGAGGAAGT
> m6A-*Irf7*-F2: GGCAAGACTTGTCAGCAGGG
> m6A-*Irf7*-R2: TAGACAAGCACAAGCCGAGAC

m6A sites on RV-EW RNA were predicted on the http://www.cuilab.cn/sramp website, and m6A-RIP-qPCR primers were designed on NCBI primer blast according to the predicted m6A sites.

m6A-RIP-qPCR primer

RIP-EW-VP1-F: ACGAAATGCTTGTTGCTATGAGT
RIP-EW-VP1-R: AACCTGTCCGTCAACCATTC
RIP-EW-VP2-F: GGCCAGAACAGGCTAAACAAC
RIP-EW-VP2-R: CGCAGTTCTCTTTCGCCATTT
RIP-EW-VP3-F: CGATGACAGCACAAAAGTCGG
RIP-EW-VP3-R: CGTGTCTCTTGCGAAGTC
RIP-EW-VP4-F: TCAGCAGACGGTTGAGACTG
RIP-EW-VP4-R: GGCTGAGATGTCATCGAAGTT
RIP-EW-NSP1-F: CCTCACATCTCTGCTACATGAACT
RIP-EW-NSP1-R: TGCTGGTTGGACATGGAATGA
RIP-EW-VP6-F: CTGCACTTTTCCCAAATGCTCA
RIP-EW-VP6-R: GAGTCAATTCTAAGTGTCAGTCCG
RIP-EW-NSP3-F: CTTGACGTGGAGCAGCAAC
RIP-EW-NSP3-R: AATGTTTCAATGTCGTCCAACG
RIP-EW-NSP2-F: TCCACCACTCTAAAGAACTACTGC
RIP-EW-NSP2-R: TCCGCTGTCATGGTGGTTTC
RIP-EW-VP7-F: TCGGAACTTGCAGACTTGAT
RIP-EW-VP7-R: GCTTCGTCTGTTTGCTGGTA
RIP-EW-NSP4-F: TGCACTGACTGTTCTATTTACGA
RIP-EW-NSP4-R: GGGAAGTTCGCATTGCTAGT
RIP-EW-NSP5/6-F: GGACACCGCAAGGTCAAAAA
RIP-EW-NSP5/6-R: TCGTCTGAGTCTGATTCTGCTT

## J2 immunofluorescent staining

IECs from *Mettl3ΔIEC* mice and from wild-type littermate control mice were isolated as described in the previous section. Isolated IECs were centrifuged onto glass slides and fixed with 4% paraformaldehyde for 30 min at room temperature. Subsequently, the IECS were permeabilized and blocked with PBS containing 0.1% Triton-X-100% and 5% bovine serum albumin for 1 hr at room temperature. Double-stranded RNA (dsRNA) was labeled by a mouse monoclonal antibody J2 (Scisons) for 2 hr at room temperature, followed by incubation with anti-mouse IgG Alexa Fluor 594-conjugated antibody (Invitrogen) for 1 hr, then cell nuclei were visualized with 4,6-diamidino-2-phenylindole (DAPI, Invitrogen). All fluorescence images were analyzed via confocal imaging using a Zeiss LSM880 microscope.

## Statistical analysis

Statistical analysis was performed with the GraphPad Prism 8.0 (GraphPad, Inc, USA). Experiments were independently replicated as indicated in the figure legends. Representative data are shown as means ± SEM. Quantitative data were compared using Student's t-test. In addition, correlational analysis of gene expression was conducted using linear regression analysis. p-values for every result are labeled on the figures: $p < 0.05$ was reckoned to be statistically significant (*$p < 0.05$, **$p < 0.01$, ***$p < 0.001$, ****$p < 0.0001$, NS, not significant).

## Acknowledgements

We would like to thank Hongdi Ma, Taidou Hu, Kaixin He, Yinglei Wang, Ji Hu, Anlei Wang, and Meng Guo for technical help and helpful discussion. This work was supported by grants from the Strategic Priority Research Program of the Chinese Academy of Sciences (XDB29030101 to SZ), the National Key R&D Program of China (2018YFA0508000 to SZ), the National Natural Science Foundation of China (81822021, 91842105, 31770990, 82061148013 and 81821001, all to SZ), and the Anhui Provincial Natural Science Foundation (2208085MC41 to WT).

## Additional information

### Funding

| Funder | Grant reference number | Author |
|---|---|---|
| Chinese Academy of Sciences | Strategic Priority Research Program (XDB29030101) | Shu Zhu |
| National Key Research and Development Program of China | 2018YFA0508000 | Shu Zhu |
| National Natural Science Foundation of China | 81822021 | Shu Zhu |
| National Natural Science Foundation of China | 91842105 | Shu Zhu |
| National Natural Science Foundation of China | 31770990 | Shu Zhu |
| National Natural Science Foundation of China | 82061148013 | Shu Zhu |
| National Natural Science Foundation of China | 81821001 | Shu Zhu |
| Anhui Provincial Natural Science Foundation | 2208085MC41 | Wanyin Tao |

The funders had no role in study design, data collection and interpretation, or the decision to submit the work for publication.

### Author contributions

Anmin Wang, Data curation, Writing – original draft, Formal analysis, Investigation, Methodology, Software, Validation, Visualization, Conceptualization; Wanyin Tao, Data curation, Investigation, Methodology, Software, Supervision, Writing – review and editing; Jiyu Tong, Data curation, Methodology, Resources; Juanzi Gao, Decai Wang, Xingxing Ren, Kaiguang Zhang, Investigation; Jinghao Wang, Chen Qian, Data curation; Gaopeng Hou, Investigation, Data curation, Formal analysis, Methodology, Resources; Guorong Zhang, Investigation, Methodology, Resources; Runzhi Li, Methodology; Siyuan Ding, Writing – review and editing, Methodology; Richard A Flavell, Resources; Huabing Li, Conceptualization, Resources; Wen Pan, Project administration, Resources, Funding acquisition, Writing – review and editing; Shu Zhu, Conceptualization, Project administration, Resources, Writing – review and editing, Supervision, Funding acquisition, Methodology

### Author ORCIDs

Anmin Wang ⓘ http://orcid.org/0000-0003-2239-9051
Richard A Flavell ⓘ http://orcid.org/0000-0003-4461-0778
Shu Zhu ⓘ http://orcid.org/0000-0002-8163-0869

### Ethics

All animal studies were performed according to approved protocols by the Ethics Committee at the University of Science and Technology of China (USTCACUC202101016).

### Decision letter and Author response

Decision letter https://doi.org/10.7554/eLife.73628.sa1
Author response https://doi.org/10.7554/eLife.73628.sa2

## Additional files

### Supplementary files
Transparent reporting form

Source data 1. Source data for *Figures 1–4*, *Figure 1—figure supplements 1–4*, *Figure 2—figure supplements 1–5*, *Figure 4—figure supplements 1–4*.

## Data availability

RNA sequencing data are available from the SRA database with accession numbers PRJNA713535. All data generated or analysed during this study are included in the manuscript and supporting file; source data files are in Dryad. Source data contain the numerical data used to generate the figures.

The following datasets were generated:

| Author(s) | Year | Dataset title | Dataset URL | Database and Identifier |
|---|---|---|---|---|
| Wang A | 2022 | Data from: m6A modifications regulate intestinal immunity and rotavirus infection | http://dx.doi.org/10.5061/dryad.p2ngf1vr8 | Dryad Digital Repository, 10.5061/dryad.p2ngf1vr8 |
| Wang A | 2022 | m6A modifications regulate intestinal immunity and rotavirus infection | https://www.ncbi.nlm.nih.gov/sra/query/acc.cgi?accPRJNA713535 | NCBI Sequence Read Archive, PRJNA713535 |

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
