## [Editor Report]

This study clearly shows that m6A modification on mRNA regulates immune responses in the intestine during rotavirus infection. The authors further show the mechanisms how m6A modification is regulated in the intestine. Thus, this study provides important insights into the regulation of anti-viral immunity in the intestine.

---

## [Decision Letter]

**Decision letter after peer review:**

Thank you for submitting your article "m6A modifications regulate intestinal immunity and rotavirus infection" for consideration by *eLife*. Your article has been reviewed by 3 peer reviewers, one of whom is a member of our Board of Reviewing Editors, and the evaluation has been overseen by Tadatsugu Taniguchi as the Senior Editor. The reviewers have opted to remain anonymous.

Essential revisions:

1) The m6A dot blot used in Figure 1 is not a good measurement system of total m6A modification levels, because the antibody used here also detects other RNA modification, m6Am (PMID: 31676230). Therefore, it is unclear if the increase of m6A dot blot intensity is due to the increase of m6A in RNAs mediated by METTL3 in IECs. The authors should investigate the m6A levels in IECs, not BMDMs, under METTL3 deficiency. Ideally, this analysis should be done using mass spectrometry.

2) The authors show that *Alkbh5* expression is increased when the mice grow up to 3 weeks old. However, the ALKBH5 protein expression changes are missing.

3) The decline in m6A levels during early life development is quite interesting. (Line 110).

– It is not clear if the subtle increase in *Alkbh5* mRNA leads to the change in global m6A levels. The authors can use ALKBH5-deficient mouse cells to confirm this point.

– Is this related to microbiota changes during the same period of life (PMID: 32165618)?

4) The authors should describe the overall phenotype of IEC-specific METTL3-deficient mice at the steady state. It is important to clarify if the augmented expression of ISG upon METTL3 deficiency is dependent on rotavirus infection. Also, the authors should describe any detectable abnormalities or changes without stimulation.

5) The finding that *Irf7* is targeted by METTL3 is not convincing.

– The authors performed MeRIP-seq and -qPCR experiments only using RNAs from wild-type IECs not from METTL3-deficient cells. It is necessary to show that the modification levels on *Irf7* mRNA is indeed reduced upon METTL3 deficiency.

– It is unclear if MeRIP-seq is properly performed or not, because there is no quality checking figure shown. For instance, the authors can generate metagene plots or gene logos of m6A modified sites to see if there is any consistency with previous reports.

– In Figure 2h, the authors should show that the change in luciferase activity between wild-type and mutant *Irf7*-3'UTR reporters is dependent on METTL3 activity by performing METTL3 knockdown or knockout. The authors should also describe how they mutagenize the sequences for clarification.

– In Figures 2F and 3C, they showed that *Irf7* is upregulated in METTL3-deficient IECs while in Figure 3F, *Irf7* is conversely downregulated in METTL3-deficient IECs. This is apparently contradictory to each other.

– As the authors mention in their discussion, m6A modification of RV RNA, endogenous retroviral elements, and IFN transcripts can all contribute to the decreased anti-viral response conferred by this modification. Can the authors provide stronger evidence ruling out these possibilities? IRF7 is an ISG itself and could mediate these other mechanisms.

6) It is unclear if the augmented expression of IRF7 per se upregulates IFN and ISG expression. Since IRF7 exerts its transcriptional activity upon phosphorylation, the authors should examine IRF7 phosphorylation and total protein levels in METTL3-deficient IECs.

7) In Figure 3, the authors utilized METTL3 and IRF7 deficient mice to show the contribution of METTL3-mediated IRF7 regulation in rotavirus infection.

– If IRF7 is totally abrogated, IFN production should be greatly impaired as shown in Figure 3A. Thus, it is not surprising to see that the IFN response is diminished. The authors can use heterozygous IRF7 deficient mice instead to check if upregulation of IRF7 under METTL3 deficiency is critical to control rotavirus infection.

– IRF7 deficient mice did not show increased susceptibility to RV, which is against previous publications (PMID: 30934842, PMID: 17301153). Can the authors explain why? One possibility is that the RV used here also shares the same NSP1 activity as SA11-4F NSP1 to directly target IRF7 for degradation (PMID: 17301153).

8) Given no effect of ALKBH5 knockout on rotavirus infection as shown in Figure 4, it is questionable if ALKBH5 has a profound role in the regulation of m6A in IECs.

– The authors should determine if m6A modification levels are increased in IECs under ALKBH5 deficiency.

– Is it possible that the other m6A eraser FTO has a redundant role? Would it be possible to use in vitro experiments to show that overexpressing ALKBH5 overcomes this immune evasion mechanism?

*Reviewer #1 (Recommendations for the authors):*

1) There are four listed corresponding authors in line 5, but two of them are not listed in line 29.

*Reviewer #2 (Recommendations for the authors):*

This manuscript sets out to determine the previously uncharacterized relationship between rotavirus (RV) infection and m6A modification during intestinal infection by using a collection of both in vivo and in vitro models. The authors found m6A modification inversely correlates with resistance to RV. The increased resistance to RV is due to elevated anti-viral IFN responses in a host with diminished m6A. They investigate mechanism by RNA-Seq and m6A-RIP-Seq and identified IRF7 as the proposed major mediator of this anti-viral response in the m6A deficient setting. During RV infection, the virus up-regulates m6A modifications to limit the IFN response, potentially targeting m6A eraser ALKBH5 for degradation by viral NSP1 protease.

The primary finding of this manuscript is novel and intriguing, as it broadens the role of this important type of RNA modification. However, there are concerns as to whether the mechanistic conclusions are sufficiently supported, and the manuscript may benefit from adding depth to certain aspects of the study prior to publication. These are detailed below.

Specific concerns:

1. Given the central role of IRF7 in the interferon response, it is perhaps not surprising that removing IRF7 reverses the resistance to RV displayed by *Mettl3* mutant mice. As the authors mention in their discussion, m6A modification of RV RNA, endogenous retroviral elements, and IFN transcripts can all contribute to the decreased anti-viral response conferred by this modification. Can the authors provide stronger evidence ruling out these possibilities? IRF7 is an ISG itself and could mediate these other mechanisms.

2. The authors suggest the reason knocking out the eraser ALKBH5 has no effect is because it is downregulated by RV NSP1. Is it possible that the other m6A eraser FTO has a redundant role? Would it be possible to use in vitro experiments to show that overexpressing ALKBH5 overcomes this immune evasion mechanism?

3. The decline in m6A levels during early life development is quite interesting and may represent a missed opportunity to broaden the impact of this manuscript. Is this related to microbiota changes during the same period of life (PMID: 32165618)?

4. In the text line 143-144, "the 5' UTR and 3' UTR of *Irf7* in ileum IECs (Figure 2d)", do they mean Figure 2c?

5. In figure 2, it would be helpful to confirm that changes are reflected at the protein level, especially demonstrating an epithelial-specific increase in IRF7.

6. In Figure 3 *Irf7*^-/-^ mice did not show increased susceptibility to RV, which is against previous publications (PMID: 30934842, PMID: 17301153). Can the authors explain why? One possibility is that the RV used here also shares the same NSP1 activity as SA11-4F NSP1 to directly target IRF7 for degradation (PMID: 17301153).

7. In Figure 3f right panel, *Irf7* expression in *Mettl3*△IEC is lower than WT, which contradicts the data in Figure 2e, f. Figure 3f is not quoted anywhere in the manuscript.

---

## [Author Response]

Essential revisions:1) The m6A dot blot used in Figure 1 is not a good measurement system of total m6A modification levels, because the antibody used here also detects other RNA modification, m6Am (PMID: 31676230). Therefore, it is unclear if the increase of m6A dot blot intensity is due to the increase of m6A in RNAs mediated by METTL3 in IECs. The authors should investigate the m6A levels in IECs, not BMDMs, under METTL3 deficiency. Ideally, this analysis should be done using mass spectrometry.

We thank the reviewer for raising a critical point. We have tried several methods to avoid the potential non-specific detection of the previous antibody (Synaptic System, #202003) we used, which was reported to detect m6Am as well.

1. We have included Dot Blot data for m6A modification in *Mettl3*△IEC and WT IECs during RV infection by using another m6A antibody (Anti-N6-methyladenosine (m6A), Sigma-Aldrich, Cat. No. ABE572-I). (see Figure 1d, 1e)

2. We have included mass spectrometry data for m6A modification in IECs during development (see Figure 1c) or RV infection (see Figure 1—figure supplement 3a).

These data suggested m6A modifications in IECs are indeed regulated during the development or RV infection. We have included the descriptions in the text (Page 4, Line 110-126).

2) The authors show that *Alkbh5* expression is increased when the mice grow up to 3 weeks old. However, the ALKBH5 protein expression changes are missing.

We thank the reviewer for raising this point. We have included the protein expression of ALKBH5 in intestine during the development (see Figure 1—figure supplement 1). The ALKBH5 protein levels are increased in the intestine along with the age (Figure 1—figure supplement 1a, Figure 1—figure supplement 1b), which is consistent to the changes of mRNA levels of *Alkbh5* during the development (Figure 1d).

3) The decline in m6A levels during early life development is quite interesting. (Line 110).– It is not clear if the subtle increase in *Alkbh5* mRNA leads to the change in global m6A levels. The authors can use ALKBH5-deficient mouse cells to confirm this point.– Is this related to microbiota changes during the same period of life (PMID: 32165618)?4) The authors should describe the overall phenotype of IEC-specific METTL3-deficient mice at the steady state. It is important to clarify if the augmented expression of ISG upon METTL3 deficiency is dependent on rotavirus infection. Also, the authors should describe any detectable abnormalities or changes without stimulation.

We actually collaborated another group and found there is a defect in intestinal stem cells in IEC-specific METTL3-deficient mice. However, as RV normally infected IECs in the villi but not in the crypt, and stem cells are not the major producers of IFN/ISGs (Sue
E.
Crawford et al., Nature
reviews
disease primers, 2017). The defect in intestinal stem cells will less likely affect the RV infection phenotype. As it is another story that are under review, we tend to not include this part of the data in our manuscript. Moreover, we have crossed *Irf7*^−/−^ mice to *Mettl3*ΔIEC mice and verified *Irf7* mediated induction of ISGs is critical for the anti-viral phenotype in *Mettl3*ΔIEC mice.

Our bulk RNA-seq data in IECs showed the augmented expression of ISGs upon METTL3 deficiency in steady state (Figure 2a). We also found an augmented ISG expression in intestine of METTL3-deficient mice in steady state or early infection of RV (2d) by qPCR. However, as the RV loads in METTL3-deficient mice during the late infection stage are significantly lower than WT mice, thus the inducible ISGs expressions are consequently lower in intestine of METTL3-deficient mice than WT mice in day 4 post infection (Figure 3f).

5) The finding that *Irf7* is targeted by METTL3 is not convincing.– The authors performed MeRIP-seq and -qPCR experiments only using RNAs from wild-type IECs not from METTL3-deficient cells. It is necessary to show that the modification levels on *Irf7* mRNA is indeed reduced upon METTL3 deficiency.– It is unclear if MeRIP-seq is properly performed or not, because there is no quality checking figure shown. For instance, the authors can generate metagene plots or gene logos of m6A modified sites to see if there is any consistency with previous reports.– In Figure 2h, the authors should show that the change in luciferase activity between wild-type and mutant *Irf7*-3'UTR reporters is dependent on METTL3 activity by performing METTL3 knockdown or knockout. The authors should also describe how they mutagenize the sequences for clarification.– In Figures 2F and 3C, they showed that *Irf7* is upregulated in METTL3-deficient IECs while in Figure 3F, *Irf7* is conversely downregulated in METTL3-deficient IECs. This is apparently contradictory to each other.

We appreciate the valuable suggestion provided by the reviewer to improve our manuscript.

We have done RIP-qPCR in METTL3 knock-down and WT MODE-K cells to verify the m6A modification on *Irf7* mRNA, the modification levels on *Irf7* mRNA is indeed reduced upon METTL3 deficiency (see Figure 2—figure supplement 1c, Figure 2—figure supplement 1d). We have added the description of the experiment in the manuscript (Page 6, Line 158-162).

We have performed metagene plots as suggested. As shown in Figure 2—figure supplement 1b, the m6A peak is enriched near the stop codon and 3’UTR region, which is consistent with previously study (Supporting Figure 1). We have added the description in the manuscript (Page 5, Line 149-151).

We have performed the luciferase assay in WT and METTL3 knock-down 293t cell, and found increased luciferase activity in mutant *Irf7*-3'UTR reporters is dependent on METTL3 activity (see Figure 2h, Figure 2—figure supplement 1e). We have added the description of the experiment into the manuscript (Page 19, Line 563-569).

IRF7 is an ISG. The expression of IRF7 is controlled by both PAMP (such as virus component)-induced transcription and post-transcriptional regulation like m6A modification mediated mRNA decay. In steady state or early stage (2d) of rotavirus infection, there is no virus or the viral loads is comparable in both *Mettl3*△IEC mice and WT mice, thus, IRF7 expression is mainly regulated by m6A and is higher in IECs from *Mettl3*△IEC mice in comparison with that from WT mice. However, as the RV loads in *Mettl3*△IEC mice during the late infection stage are significantly lower than WT mice, in this case, IRF7 expression is mainly regulated by the PAMP from virus, thus the inducible IRF7 expressions is consequently lower in intestine of *Mettl3*△IEC than WT mice in day 4 post infection (Figure 3f).

– As the authors mention in their discussion, m6A modification of RV RNA, endogenous retroviral elements, and IFN transcripts can all contribute to the decreased anti-viral response conferred by this modification. Can the authors provide stronger evidence ruling out these possibilities? IRF7 is an ISG itself and could mediate these other mechanisms.6) It is unclear if the augmented expression of IRF7 per se upregulates IFN and ISG expression. Since IRF7 exerts its transcriptional activity upon phosphorylation, the authors should examine IRF7 phosphorylation and total protein levels in METTL3-deficient IECs.

We have provided the phosphorylation and total protein levels of IRF7 and TBK1 in MODE-K cells treated with poly I:C. Both total IRF7 and phosphorylated IRF7 are upregulated in METTL3 knock down cells compare to control cells (see Figure 2—figure supplement 1f). However, Both total TBK1 and phosphorylated TBK1 remain unchanged (Figure 2—figure supplement 1f), suggesting the augmented ISGs are less likely due to the activation of the upstream signal of IFN.

7) In Figure 3, the authors utilized METTL3 and IRF7 deficient mice to show the contribution of METTL3-mediated IRF7 regulation in rotavirus infection.– If IRF7 is totally abrogated, IFN production should be greatly impaired as shown in Figure 3A. Thus, it is not surprising to see that the IFN response is diminished. The authors can use heterozygous IRF7 deficient mice instead to check if upregulation of IRF7 under METTL3 deficiency is critical to control rotavirus infection.

We thank the reviewer for pointing out an important issue. However, we checked the IRF7 expression levels in IECs from *Irf7*^+/+^, *Irf7*^+/-^ and *Irf7*^-/-^ mice and found that there is no difference between IRF7 levels in IECs from *Irf7*^+/-^ mice and that in IECs from *Irf7*^+/+^ mice. Thus, it is not feasible to use heterozygous IRF7 deficient mice to test the idea (Supporting Figure 2).

6) Given no effect of ALKBH5 knockout on rotavirus infection as shown in Figure 4, it is questionable if ALKBH5 has a profound role in the regulation of m6A in IECs.– The authors should determine if m6A modification levels are increased in IECs under ALKBH5 deficiency.

We performed the m6A dot blot assay to detect m6A modification levels in ALKBH5-knock down MODE-K cells and we do find an increase of m6A modification level under ALKBH5 deficiency (see Figure 4—figure supplement 1). No effect of ALKBH5 knockout on rotavirus infection actually puzzled us as well before (Figure4c, 4d and 4e), until we found RV infection down-regulated ALKBH5 expression in the intestine of WT mice (Figure4a).

Reviewer #1 (Recommendations for the authors):1) There are four listed corresponding authors in line 5, but two of them are not listed in line 29.

We apologize for the possible confusion. According to the contribution made by each author, we have negotiated and selected two as corresponding authors.

Reviewer #2 (Recommendations for the authors):This manuscript sets out to determine the previously uncharacterized relationship between rotavirus (RV) infection and m6A modification during intestinal infection by using a collection of both in vivo and in vitro models. The authors found m6A modification inversely correlates with resistance to RV. The increased resistance to RV is due to elevated anti-viral IFN responses in a host with diminished m6A. They investigate mechanism by RNA-Seq and m6A-RIP-Seq and identified IRF7 as the proposed major mediator of this anti-viral response in the m6A deficient setting. During RV infection, the virus up-regulates m6A modifications to limit the IFN response, potentially targeting m6A eraser ALKBH5 for degradation by viral NSP1 protease.The primary finding of this manuscript is novel and intriguing, as it broadens the role of this important type of RNA modification. However, there are concerns as to whether the mechanistic conclusions are sufficiently supported, and the manuscript may benefit from adding depth to certain aspects of the study prior to publication. These are detailed below.

We appreciate the supportive comments and the valuable suggestion of new experiments provided by the reviewer to improve our manuscript. We believe that the new data included in the revised manuscript further validate and support our claim.

Specific concerns:1. Given the central role of IRF7 in the interferon response, it is perhaps not surprising that removing IRF7 reverses the resistance to RV displayed by *Mettl3* mutant mice. As the authors mention in their discussion, m6A modification of RV RNA, endogenous retroviral elements, and IFN transcripts can all contribute to the decreased anti-viral response conferred by this modification. Can the authors provide stronger evidence ruling out these possibilities? IRF7 is an ISG itself and could mediate these other mechanisms.

We thank the reviewer for raising an important point. We constructed plasmids pLVX-*Irf7*-WT (*Irf7*-WT) or pLVX-*Irf7*-Mut containing mutated m6A modification sites (*Irf7*-MUT) and transduced it into *Irf7*^-/-^ MEF cells. We found that *Irf7* without m6A modification led to increased expression of *Irf7* and ISGs (Figure 2—figure supplement 4a and 4b), and enhanced antiviral responses in MEF cells (Figure 2—figure supplement 4a). These data cannot completely exclude other mechanisms but further highlighted the involvement of m6A modification on *Irf7* in the resistance phenotype to virus infection by METTL3 deficiency.

2. The authors suggest the reason knocking out the eraser ALKBH5 has no effect is because it is downregulated by RV NSP1. Is it possible that the other m6A eraser FTO has a redundant role? Would it be possible to use in vitro experiments to show that overexpressing ALKBH5 overcomes this immune evasion mechanism?

We thank the reviewer for the comments. We checked the expression of *Alkbh5/ALKBH5* and *Fto/FTO* in our RNA-seq data in mouse IECs, and also a previous reported RNA-seq data in human intestinal enteroid (Kapil Saxena, et al., PNAS. 2017). We found the *Alkbh5/ALKBH5* expression is much higher than *Fto/FTO* in intestine (Figure 4—figure supplement 1a). We did qPCR analysis in mouse epithelial cell line MODE-K and showed the similar results (Figure 4—figure supplement 1a). Furthermore, we knock down METTL3, ALKBH5 and FTO in MODE-K cell. Through dot blot assay, we found ALKBH5 but not FTO is the dominant m6A eraser in IECs (Figure 4—figure supplement 1).

Then we transduced plasmid encoding ALKBH5 or control plasmid in MODE-K cell and infected these cells with RRV. Indeed, we found the overexpression of ALKBH5 reinforced the protein level of ALKBH5, increase the IFN/ISGs expression and inhibit RRV infection in MODE-K cell (Figure 4—figure supplement 2), indicating overexpressing ALKBH5 overcomes the immune evasion mechanism mediated by RV.

3. The decline in m6A levels during early life development is quite interesting and may represent a missed opportunity to broaden the impact of this manuscript. Is this related to microbiota changes during the same period of life (PMID: 32165618)?

We thank the reviewer for the positive comment. To address the question whether the decline in m6A modifications is related to microbiota development during early life development, we examined the m6A modifications in ileum tissue from germ free mice at 1 week or 3 weeks of age. There is no difference between these two groups, which indicated microbiota changes may contribute to the decline in m6A levels during early life development (Figure 1—figure supplement 2).

4. In the text line 143-144, "the 5' UTR and 3' UTR of *Irf7* in ileum IECs (Figure 2d)", do they mean Figure 2c?

We thank the reviewer for pointing this out, we have corrected it in the manuscript.

5. In figure 2, it would be helpful to confirm that changes are reflected at the protein level, especially demonstrating an epithelial-specific increase in IRF7.

We have provided the phosphorylation and total protein levels of IRF7 and TBK1 in intestinal epithelial MODE-K cells treated with poly I:C. Both total IRF7 and phosphorylated IRF7 are upregulated in METTL3 knock down cells compare to control cells (see Figure 2—figure supplement 1f). However, Both total TBK1 and phosphorylated TBK1 remain unchanged (Figure 2—figure supplement 1f), suggesting the augmented ISGs are less likely due to the activation of the upstream signal of IFN.

6. In Figure 3 *Irf7*^-/-^ mice did not show increased susceptibility to RV, which is against previous publications (PMID: 30934842, PMID: 17301153). Can the authors explain why? One possibility is that the RV used here also shares the same NSP1 activity as SA11-4F NSP1 to directly target IRF7 for degradation (PMID: 17301153).

We thank the reviewer for raising a interesting point. We have reported that RV RNA methyl- and guanylyl-transferase (VP3) can degrade MAVS, an adaptor upstream of IRF7, to evade the host immune response towards RV in a host-range-restricted manner (Siyuan
Ding et al., *eLife*, 2018). In the study (Siyuan
Ding et al., *eLife*, 2018), we showed *Mavs*^-/-^ also did not show increased susceptibility to RV, which is the same we observed in *Irf7*^-/-^ mice in this manuscript (Wang et. al). Thus, we hypothesize that the rotavirus EW strain can degrade MAVS and thus inhibit the MAVS/IFN/IRF7 signaling in mice, which led to no enhanced susceptibility to RV in either *Mavs*^-/-^ or *Irf7*^-/-^ mice.

7. In Figure 3f right panel, *Irf7* expression in *Mettl3*△IEC is lower than WT, which contradicts the data in Figure 2e, f. Figure 3f is not quoted anywhere in the manuscript.

IRF7 is an ISG. The expression of IRF7 is controlled by both PAMP (such as virus component)-induced transcription and post-transcriptional regulation like m6A modification mediated mRNA decay. In steady state or early stage (2d) of rotavirus infection, there is no virus or the viral loads is comparable in both *Mettl3*△IEC mice and WT mice, thus, IRF7 expression is mainly regulated by m6A and is higher in IECs from *Mettl3*△IEC mice in comparison with that from WT mice. However, as the RV loads in *Mettl3*△IEC mice during the late infection stage are significantly lower than WT mice, in this case, IRF7 expression is mainly regulated by the PAMP from virus, thus the inducible IRF7 expressions is consequently lower in intestine of *Mettl3*△IEC than WT mice in day 4 post infection (Figure 3f).

Figure 3f showed the expression of *Mettl3* or *Irf7* to indicate the successful knockouts of the genes. We have quoted in the manuscript (Page 8, line 220).